# Macromutations Yielding Karyotype Alterations (and the Process(es) behind Them) Are the Favored Route of Carcinogenesis and Speciation

**DOI:** 10.3390/cancers16030554

**Published:** 2024-01-28

**Authors:** Ingo Schubert

**Affiliations:** Leibniz Institute of Plant Genetics and Crop Plant Research (IPK), 04644 Gatersleben, Germany; schubert@ipk-gatersleben.de

**Keywords:** chromosome mutations, DNA double-strand break repair, karyotype, meiosis, evolution, carcinogenesis, speciation

## Abstract

**Simple Summary:**

The evolutionary events, carcinogenesis and speciation, are presumably based on a phase of ‘genome instability’ caused by DNA breakage and mis-repair leading to chromosome rearrangement. Subsequently, the corresponding initial cells pass bottlenecks of selection, yielding either a tumor or an organism with a balanced genome, structurally different from the ancestral one.

**Abstract:**

It is argued that carcinogenesis and speciation are evolutionary events which are based on changes in the ‘karyotypic code’ through a phase of ‘genome instability’, followed by a bottleneck of selection for the viability and adaptability of the initial cells. Genomic (i.e., chromosomal) instability is caused by (massive) DNA breakage and the subsequent mis-repair of DNA double-strand breaks (DSBs) resulting in various chromosome rearrangements. Potential tumor cells are selected for rapid somatic proliferation. Cells eventually yielding a novel species need not only to be viable and proliferation proficient, but also to have a balanced genome which, after passing meiosis as another bottleneck and fusing with an identical gamete, can result in a well-adapted organism. Such new organisms should be genetically or geographically isolated from the ancestral population and possess or develop an at least partial sexual barrier.

A long-lasting discussion refers to whether speciation is based on the gradual accumulation of ‘micromutations’ (gene mutations), or rather on saltatory ‘macromutations’ (chromosome rearrangements which change the order of sequences) (for a recent reviews, see [1,2,3]). Among the various alternative hypotheses and explanations, the punctuated equilibrium theory, proposed by Niles Eldredge and Stephen Jay Gould about thirty years ago, has been the most influential one [4]. According to this theory, evolution should be characterized by long periods of relative stability (stasis), punctuated by brief episodes of rapid change. Unfortunately, due to difficulties in proposing the mechanisms of both stasis and punctuated fossil changes under neo-Darwinism’s gene-centered framework, i.e., assuming that micro- and macroevolution are both based on gene mutations, Gould later retreated from this vital concept [5]. In the following, the arguments for either option will be compared based on recent advances in cancer evolutionary research, which provides both a new experimental platform and a genome-based evolutionary perspective. Observations about tumor development led to the idea of a ‘two-step-evolution’ of cancer, with the first step being a chaotic event, and the second one a selection step through a bottleneck of genome instability. The chaotic event leads to genome instability via massive chromosome rearrangements, after which only a small minority of cells with the ability to proliferate quickly and to escape from immune surveillance can be rescued from ‘genomic chaos’ and yield tumors with diverse structural and/or numerical chromosome mutations. In fact, karyotype alterations are universally observed in the vast majority of cancer cases [2]. Furthermore, massive karyotype changes in cancer are well documented in the literature following the cancer genome sequencing project [6,7,8]. In principle, the same scheme can also apply to speciation. Environmental disasters resulting in mass extinctions—for instance, of dinosaurs ~66 million years ago—provided reduced competition and nearly empty ecological niches. This offered the opportunity for the fast radiation of mammals via adaptation, mediated by novel genetic variants (for review see [2,9]). Also, under less dramatic circumstances, events causing multiple chromosome breakages and rearrangements in one or few individuals could initiate speciation. The observation that even related species differ in their genome arrangement, i.e., in their karyotypes, favors this assumption.

Usually, **genome chaos occurs after (multiple) breakages of genomic DNA**, elicited by external, e.g., ionizing irradiation or other genotoxin exposure (as in drug-mediated cancer therapy), or by endogenous impacts. Endogenous impacts include, for example, the mechanical rupture of anaphase bridges formed by dicentric chromosomes and result in mitotic bridge-breakage fusion cycles [10] that yield complex rearrangements (Figure 1). Alternatively, micronuclei could be endonucleolytically degraded, and, after fragment tethering and their re-integration into one daughter nucleus, may lead to massive rearrangements without an essential loss of genetic information (chromothripsis) [11,12,13]. Micronuclei harbor mis-segregated chromosomes (mainly acentric chromosome fragments), wrapped by a nuclear membrane during the exit from mitosis.

**Broken DNA fragments may trigger various pathways of DNA double-strand break (DSB) repair** (see Figure 2). The different DSB repair pathways can either precisely restore the pre-break situation or erroneously lead to deletions, insertions, or, via the ligation of break ends different to the original ones, to chromosome rearrangements (see Figure 2 and Figure 3). A repair bias towards deletions results in genome shrinkage. A bias towards insertion causes genome expansion (including the spreading of retroelements). Such repair biases can also explain the so-called genome-size paradox (=”C-value paradox” [15]), according to which the genome size is not correlated with the genetic complexity of organisms [16]. A switch towards or between these biases may occur via mutations in individual components of the protein complexes involved in DSB repair [17]. Ligating ends from different breaks rearrange the genome, leading to inversions, translocations, or transpositions (Figure 3). The more break ends that are present simultaneously, the lower the chance of ‘correct’ repair, which restores the pre-break situation. Instead, any types of ‘mis-repair’ yielding sequence alterations at a genic or karyotype level, i.e., deletions, insertions, or other rearrangements, become abundant. Comparable to cancer evolution, after the (mis-)repair of DSBs, only a single or few cells may pass the bottleneck of selection. Independent of the type of the respective chromosome aberrations, the harboring cells have to be genetically balanced and able to proliferate to eventually reach meiosis via the germline in animals, or via the apical meristem in plants.

**Meiosis is another bottleneck for cells with rearranged karyotypes**. Through meiotic crossover and the random segregation of paternal and maternal chromosomes, parental alleles become stirred into a new combination. True genetic novelty, however, is exclusively obtained via mutagenesis. Among mutations, ‘macromutations’, as consequences of DSB mis-repair, alter the arrangement of genetic information along chromosomes (large duplications, deletions, inversions, transpositions, as well as translocations). Such rearrangements are usually counter-selected during meiosis. This happens because the homologous recombination repair of Spo11-induced DSBs, involving paired homologous chromosomes, may be disturbed between structurally heterozygous homologs. Even more importantly, structurally heterozygous chromosomes form multivalent pairing configurations during meiosis I and may (frequently) lead, via mis-segregation, to imbalanced gametes (Figure 4). Only if altered chromosome complements are genetically balanced, and gametes carry the same alteration fuse, will the resulting homozygous organism, in turn, perform correct meiotic division. Backcrossing with the ‘wildtype’ would again cause meiotic disturbance, yield a reduced number of fertile gametes, and thus result in less progeny. Therefore, meiosis assures the stable inheritance of genetic material, discriminating against structural chromosome variations, as shown experimentally in yeast [19]. In this way, meiosis leads, in general at least, to the genetic isolation of individuals with different karyotypes [20] because karyotypes preserve the order of genes along the chromosome. This species-specific order of genes organizes the genetic network of a given species, as the topological relationships among different sequences on the chromosome represent system-level information beyond the genes.

**Mis-repair of DSBs** in proliferation-proficient somatic cells **generates genetic novelty**, which as genome-rearranging macromutations may lead to evolutionarily new karyotypes [21] and form a (more or less strict) sexual barrier. Whether the new karyotypic information, the ‘karyotype code’, with an altered sequence order [22,23] results directly in a new species, which then gradually accumulates genic micromutations and epigenetic alterations, i.e., morphological and/or physiological peculiarities, depends on the degree of hybrid incompatibility (under sympatric conditions), and/or on the duration of the geographic isolation of the new variant. In general, hybrid incompatibility should increase with the size and number of genome rearrangements. Smaller inversions, duplications, or transpositions do not necessarily disturb meiotic processes. For multiple chromosome rearrangements—for instance, those which distinguish the karyotypes of the sister duckweed species, *Spirodela polyrhiza* and *S. intermedia* [24,25]—it is likely that many, if not all, occurred simultaneously, via a chaotic event, rather than step-by-step individually (Figure 5). In the former case, a sexual barrier would have arisen immediately. 

**New species can also result via reticulate evolution**, if the (rare) fusion of gametes of two already established species with differentiated karyotypes occurs. A prerequisite is bivalent formation and correct chromosome segregation in meiosis I of the hybrid organism. Meiotic stability is achieved if, after the fusion of reduced gametes, a chromosome doubling occurs, or if unreduced gametes fuse. In both cases, regular bivalent formation and correct segregation take place in meiosis I, enabling stable propagation of the carrier organism as a novel species. If stabile hybrids maintain the parental chromosome complements, they are called ‘neopolyploids’. If such hybrids are ‘diploidizing’ via further rearrangements within or between the parental karyotypes—often by translocations which reduce the chromosome number and genome size, e.g., via loss of centromeres and telomeres (=descending dysploidy, for review see [26])—they are called ‘mesopolyploids’. If the hybridization events are revealed only by genomic approaches, detecting multiple variants of individual genes, the corresponding species are considered as ‘paleopolyploids’, which are already largely diploidized. Stress-mediated ‘genomic shock’ [27] may be induced directly by interspecific hybridization, or by the subsequent meiotic loss of so-called ‘gametocidal chromosomes’, as occurs in some interspecific hybrids in cereals (for a review, see [28]) and appears as multiple DNA breakage. The reasons and pathways are not yet clear, but such events yield genomic chaos, i.e., genome rearrangement via DSB mis-repair, and thereby contribute to sexual isolation and speciation.

Alternatively, speciation might occasionally be initiated by one (or few) genetic mutation (e.g., in plants altering flowering time), if such mutations cause an immediate sexual barrier.

Thus, it seems evident that **cancer formation and speciation share many key features**. More fundamentally, both cases share processes of creating a new system from an old one and involve the preservation and growth of these new systems through population expansion. In particular, both exhibit a pattern of two-phased evolution with genome reorganization-mediated punctuated macroevolution, followed by gene mutation-mediated microevolution. Interestingly, these evolutionary mechanisms can be simply described as information creation (by changing the karyotype), information preservation (by largely maintaining gene synteny), information modification (via gene mutations and epigenetic alterations), and information usage (in various physiological processes). Significantly, this view on carcinogenesis and speciation links macroevolution to chromosome structural changes, while microevolution is linked to gene-level changes, and is distinct from the traditional viewpoint that equates macroevolution with either special gene mutations or with the accumulation of microevolutionary events over time. Additionally, it explains the phenomenon of massive speciation following massive extinction well, as well as the long periods of stasis in the fossil record, given that the function of sex can preserve the karyotype code. It is noteworthy that the induced genome chaos observed in cancer research bears a striking resemblance to the ‘genetic earthquake’ experiments of Barbara McClintock in corn. Her observed karyotype changes might potentially lead to the emergence of novel species. Recently, numerous large-scale comparative sequencing experiments in both plants and animals have revealed that karyotype reorganization is a universal mechanism for biodiversification (e.g., [29,30,31]). Now, with the new concepts of karyotype coding and the two-phased evolution, the time is ripe to establish the genome and information-based evolutionary framework [22,23].

The implications of structural (and/or numerical) chromosomal variability (chromosomal instability = CIN) for cancer research and treatment are profound. The molecular genetic tradition often emphasizes gene mutations in discussions about cancer, with less appreciation for the creation of a new system with an altered karyotype. Prioritizing the importance of CIN in many cancers parallels karyotype alteration towards speciation, given that cancer is an unstable system, involving both karyotypic macro- and genic microevolution similarly as speciation does. Understanding the diverse triggering factors of cancer, and its response to mutagenic treatments, is now closely tied to the level of CIN through environmental and internal mutagenic-induced stress, mediating evolutionary processes [32]. The challenge lies in the chaotic character of evolutionary processes during which the same mechanisms, such as DNA repair or meiosis, can contribute either to evolutionary innovation (by macroevolution) or to restore or (more or less) maintain the original state (during microevolution). This insight is of clinical relevance and should be integrated into treatment plans because treatment options could induce rapid genome chaos leading to swift drug resistance and increased cancer aggressiveness [33,34,35].

## Conclusions

In **conclusion**, it seems reasonable to state that speciation at a diploid level is most likely based on the genetic isolation of individuals with karyotypes differing from ancestral ones by rearrangements (macromutations) after the mis-repair of multiple DNA double-strand breaks. Thus, chromosome rearrangements are initial phenomena in most cases of speciation, as well as in cancer development.

This is why chromosome instability via DNA breakage and mis-repair is a driver of carcinogenesis and speciation, and is a genotypic as well as phenotypic feature of the initial cells for both evolutionary processes. The main difference between speciation and carcinogenesis is that cancer development is not limited by sexual constraints and thus multiple cycles of two-phased evolution may happen within one organism, proceeding much faster than speciation in multicellular eukaryotic organisms.

## Figures and Tables

**Figure 1 cancers-16-00554-f001:**
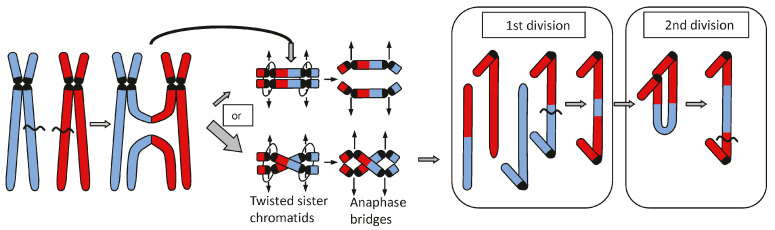
After replication of a dicentric translocation product, and a twist of its sister chromatids between centromeres, an anaphase bridge can form (left part of the figure). The chance for twisting increases with the distance of centromeres. After random rupture of the bridge and another round of replication, the break-ends of the sister chromatids ‘fuse’, yielding again a bridge and another random break in the next anaphase, mediating complex rearrangements (right part of the figure), modified after Schubert (2021) [14].

**Figure 2 cancers-16-00554-f002:**
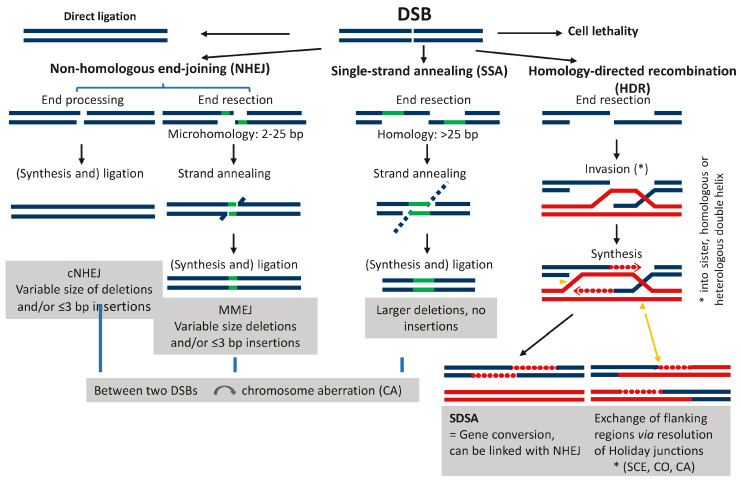
DSB repair pathways and potential molecular consequences, modified according to Vu et al. (2014) [18]. cNHEJ = classical NHEJ; MMEJ = microhomology-mediated end-joining; SDSA = synthesis-dependent strand annealing; SCE = sister chromatid exchange; CO = crossover.

**Figure 3 cancers-16-00554-f003:**
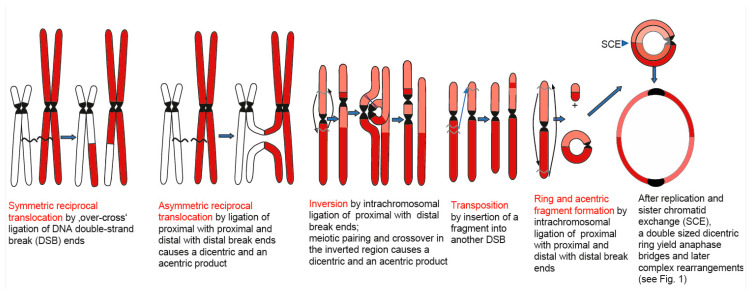
DSB repair by ligation of ends from different breaks results in various chromosome rearrangements. Dicentric chromatids resulting from asymmetric translocation, from crossover between inverted regions of homolgous chromosomes, or from dicentric rings yield anaphase bridges and subsequently complex rearrangements (see Figure 1, modified according to Schubert, 2021 [14]).

**Figure 4 cancers-16-00554-f004:**
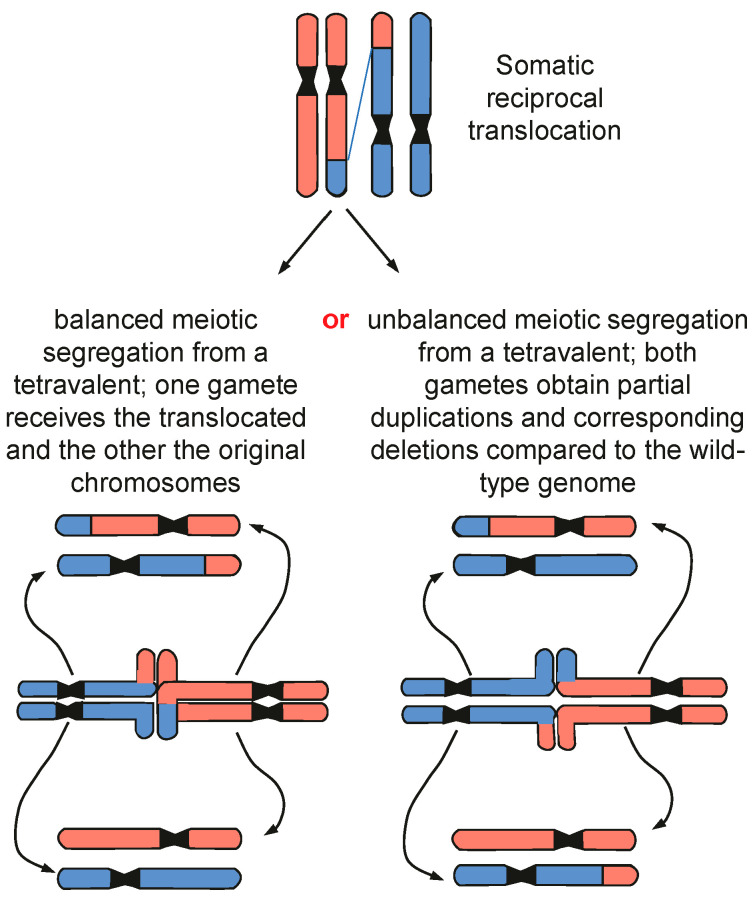
Balanced and unbalanced meiotic segregation of translocation chromosomes. Only the gametes with balanced translocation chromosomes are viable and, if fused with the same type of gamete, can lead to genetic isolation and speciation. Modified according to Schubert, 2021 [14].

**Figure 5 cancers-16-00554-f005:**
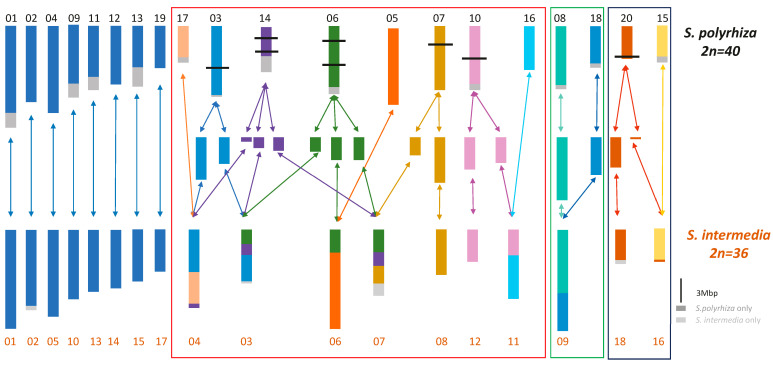
Chromosome rearrangements between the duckweed species *Spirodela polyrhiza* and *S. intermedia* as revealed by fluorescent in situ hybridisation with bacterial artificial chromosomes mapped to *S. polyrhiza* (Hoang & Schubert 2017 [24]; modified according to Hoang et al. 2022 [25]). Frames in different color indicate (groups of) rearrangements. In particular, the first complex group of rearrangements likely occurred simultaneously.

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
