# Peer review of "Macromutations Yielding Karyotype Alterations (and the Process(es) behind Them) Are the Favored Route of Carcinogenesis and Speciation"

_cancers, 2024, doi:10.3390/cancers16030554_

Round 1

Reviewer 1 Report

Comments and Suggestions for Authors

The novelty in this manuscript is not significant, and there is not much exciting work.

Comments on the Quality of English Language

Minor editing of English language required

Author Response

The reviewer is in so far right as no novel facts are presented. The novelty is just the link between cancerogenesis and speciation via genome instability, the basis of which seem to be chromosome breakage and DSB (mis)repair as explicated in some detail. According to the suggestions of referee 4, the Introduction has been expanded by including more references.

Reviewer 2 Report

Comments and Suggestions for Authors

Dr. Schubert presents an intellectually stimulating hypothesis that seeks to establish a connection between carcinogenesis and speciation by suggesting a shared foundation in genomic instability. This instability, with its DNA breakage and subsequent faulty repair leading to chromosomal rearrangements, is well-acknowledged in cancer biology. Its proposed link to speciation is a noteworthy concept that warrants further exploration. However, the presentation of this hypothesis presumes a considerable degree of prior knowledge, which could make the material less accessible to a broader audience. My primary critique is that the paper would benefit greatly from additional explanatory content or visual aids that delineate the mechanisms of tumor evolution and speciation more clearly. This enhancement would not only make the complex concepts more digestible but also strengthen the overall impact of the hypothesis.

Author Response

I am not sure what the reviewer had in mind when requesting "additional explanatory content or visual aids that delineate the mechanisms of tumor evolution and speciation". In my opinion, the 5 figures illustrate the joint mechanisms (chromosome breakage and DSB (mis)repair) as basic processes for cancerogenesis and speciation.

Reviewer 3 Report

Comments and Suggestions for Authors

The argument presented here outlines a fascinating perspective on the interconnected processes of carcinogenesis (cancer development) and speciation (formation of new species) within an evolutionary framework. Here are some comments on the key points:

1.      The concept of linking carcinogenesis and speciation as evolutionary events provides an intriguing view, suggesting common underlying mechanisms.

2.      Highlighting changes in the "karyotypic code" underscores the significance of chromosomal alterations in driving evolutionary processes.

3.      The emphasis on "genome instability" as a trigger for these events aligns with the understanding that genetic changes play a pivotal role in evolution.

4.      The mention of massive DNA breakage leading to chromosome rearrangements sheds light on the molecular mechanisms involved, adding a layer of detail to the argument.

5.      The notion of selecting potential tumor cells for rapid somatic proliferation introduces the concept of natural selection operating at the cellular level, emphasizing the importance of viability and adaptability.

6.      The requirement for cells yielding a novel species to have a balanced genome, especially after passing through meiosis, reflects the complexity of genetic factors influencing speciation.

In conclusion, this argument intertwines genetic and evolutionary concepts, offering a different perspective on the possible interplay between cancer development and the emergence of new species. The intricate balance of genetic factors and selective pressures highlighted in the narrative contributes to a comprehensive understanding of these evolutionary phenomena.

Comments on the Quality of English Language

The quality of English language in the provided text is generally good. The sentences are well-structured, and the vocabulary used is appropriate for discussing scientific concepts.

Author Response

Thank you for kind comments. Based on the suggestions of referee 4, a few remarks and references that support the statements of the manuscript have now been added.

Reviewer 4 Report

Comments and Suggestions for Authors

Manuscript entitled "Macromutations yielding karyotype alterations (and the process(es) behind) are the favored route of carcinogenesis and speciation"

Major issues:

1. The authors should provide some representative karyotyping as illustrations for their story.

2. The authors sould perform review on the published papers related to karyotyping in selective cancer types to support their story. 

3. The authors should try to gain more clinical relevance.

Comments on the Quality of English Language

acceptable.

Author Response

On p.2 and p.7 remarks and corresponding new references have been included which illustrate karyotyping in various cancers and point on the clinical relevance of karyotype changes.

Reviewer 5 Report

Comments and Suggestions for Authors

In the submitted manuscript author gave a brief exposé on the similarity of cellular processes behind speciation and carcinogenesis.

It is interesting and well written text, however, Wikipedia shout not be used as a truthful source of scientific information but only peer-reviewed papers and books. And for all on-line references a date of last accessing should be provided, as journal requires.

In addition, all abbreviations presented in Figure 3 (DSB and SCE) should be explained in figure legend, regardless their previous explanation in the main text or other figure's legend.

Author Response

The abbreviations in Fig, 3 are now spelled out. The date of last accessing to on-line references is included. Although the journal accepts Wikipedia pages as reference, an additional reference is gives now at the corresponding positions.

Round 2

Reviewer 1 Report

Comments and Suggestions for Authors

In this study the author tries to link macromutations yielding karyotype alterations with carcinogenesis and speciation which is an interesting topic to consider. But, in my opinion, the revised version is not sufficiently improved and not enough review was provided on karyotyping linked with cancer types and need to provide more typical karyotyping which is still a limitation of this study. This improvement will strengthen the overall impact of this study.

Author Response

References of several actual papers regarding karyotyping of cancers have been added during revision and, of course, there are too many to cite them all. This would also bring the references out of balance with regard to the scope of the article. If the reviewer has one or a few in mind which are considered of particular importance for the phenomenom, they should please be mentioned in the reviewers comments.

Reviewer 4 Report

Comments and Suggestions for Authors

The revision is acceptable.

Comments on the Quality of English Language

The revision is acceptable

Author Response

Thank you for positve remarks